# A universal pocket in fatty acyl-AMP ligases ensures redirection of fatty acid pool away from coenzyme A-based activation

Gajanan S Patil[1,2†], Priyadarshan Kinatukara[1†], Sudipta Mondal[1], Sakshi Shambhavi[1,2], Ketan D Patel[1], Surabhi Pramanik[1], Noopur Dubey[1], Subhash Narasimhan[1], Murali Krishna Madduri[1], Biswajit Pal[1], Rajesh S Gokhale[3], Rajan Sankaranarayanan[1,2*]

[1]CSIR-Centre for Cellular and Molecular Biology, Hyderabad, India; [2]Academy of Scientific and Innovative Research (AcSIR), Ghaziabad, India; [3]National Institute of Immunology, New Delhi, India

**\*For correspondence:**
sankar@ccmb.res.in

†These authors contributed equally to this work

**Abstract** Fatty acyl-AMP ligases (FAALs) channelize fatty acids towards biosynthesis of virulent lipids in mycobacteria and other pharmaceutically or ecologically important polyketides and lipopeptides in other microbes. They do so by bypassing the ubiquitous coenzyme A-dependent activation and rely on the acyl carrier protein-tethered 4'-phosphopantetheine (*holo*-ACP). The molecular basis of how FAALs strictly reject chemically identical and abundant acceptors like coenzyme A (CoA) and accept *holo*-ACP unlike other members of the ANL superfamily remains elusive. We show that FAALs have plugged the promiscuous canonical CoA-binding pockets and utilize highly selective alternative binding sites. These alternative pockets can distinguish adenosine 3',5'-bisphosphate-containing CoA from *holo*-ACP and thus FAALs can distinguish between CoA and *holo*-ACP. These exclusive features helped identify the omnipresence of FAAL-like proteins and their emergence in plants, fungi, and animals with unconventional domain organizations. The universal distribution of FAALs suggests that they are parallelly evolved with FACLs for ensuring a CoA-independent activation and redirection of fatty acids towards lipidic metabolites.

## Introduction

The ANL superfamily includes enzymes such as the acyl/aryl-CoA ligases (ACS or FACLs), adenylation domains (A-domains), and luciferases along with the recently identified fatty acyl-AMP ligases (FAALs) (*Schmelz and Naismith, 2009*). These enzymes are involved in the production of both primary metabolites such as acyl-CoA and secondary metabolites such as antibiotics (*Sieber and Marahiel, 2005*), complex lipids (*Gokhale et al., 2007*), cyclic peptides (*Walsh, 2004*), and lipopeptides (*Hansen et al., 2007*; *Hemmerling et al., 2018*). Basic metabolic pathways such as β-oxidation, membrane biogenesis, post-translational modifications, etc., use primary metabolites such as acyl-CoA. The secondary metabolites such as complex lipids that function as virulent molecules in *Mycobacteria* and bioactive molecules in several microbes that help tide over unfavorable conditions and establish themselves in their niches. Such diverse metabolites are produced by the members of the ANL superfamily through a two-step catalytic mechanism. It begins with the activation of carboxylate moiety containing substrates such as fatty acids or amino acids by adenosine triphosphate (ATP) hydrolysis and finally transferring it to an acceptor such as CoA or *holo*-ACP. Multiple structural and biochemical studies show that members of the superfamily such as FACLs and A-domains employ a common pocket for

the chemically identical CoA and the 4'-PPant moieties attached to the *holo*-ACP, respectively, for the final transfer. It was later demonstrated that the A-domains can cross-react with CoA to form amino-acyl-CoA (*Linne et al., 2007*), which points to the liabilities of utilizing a common pocket for binding chemically identical moieties. Promiscuity towards the final acceptor has now been noted in different classes of ANL superfamily members where luciferases are shown to catalyze fatty acyl-CoA formation (*Oba et al., 2003*) and FACLs producing bioluminescence with molecular oxygen (*Oba et al., 2009*). While fatty acid/amino acid substrate promiscuity is well studied and exploited in combinatorial biosynthesis of bioactive molecules (*Winn et al., 2020*), the origin and basis of acceptor promiscuity is relatively less understood.

FAALs are atypical enzyme systems of the ANL superfamily as these show a narrow preference for *holo*-ACP and not CoA, where they transfer the activated fatty acyl-AMP to the 4'-PPant of *holo*-ACP (*Trivedi, 2004*). FAALs rejecting the small, diffusible, and abundant CoA while accepting the ACP-tethered to a 4'-PPant moiety (*Figure 1—figure supplement 1a and b*) is puzzling as they are chemically identical. In a previous study, it was proposed that the FAAL-specific insertion (FSI), an additional stretch of amino acids found only in the N-terminal domain of FAALs, prevents reaction with CoA (*Arora, 2009*). However, the deletion or destabilization of the FSI failed to convert FAALs as efficient producers of acyl-CoA as there is only a weak ability to react with CoA (*Arora, 2009*; *Goyal et al., 2012*). Moreover, it is also unclear how such a mechanism operates and distinguishes the two chemically identical acceptors, particularly when CoA is an abundant metabolite. These observations prompted us to hypothesize that FAALs have either evolved novel appendages or other modes for binding the acceptor to allow strict rejection of CoA.

In the present study, we have used structural, mutational, and biochemical analyses to identify the mechanistic basis of how FAALs can distinguish between near identical acceptors for the acyl transfer reaction. We show that, unlike other members of the superfamily, FAALs achieve acceptor fidelity by avoiding the usage of a promiscuous canonical CoA-binding pocket and utilizing a discriminatory pocket that is distinct from the canonical CoA-binding pocket. Loss- and gain-of-function mutations were generated by identifying the structural determinants that nullify the canonical CoA-binding pocket. Interestingly, we found that the non-functional canonical CoA-binding pocket and the unique discriminatory alternative pocket are unique features of FAALs, which is conserved in all forms of life including plant, fungi, and animals. The identification of such a conserved rejection mechanism across organisms has larger implications in determining the redirection of cellular flux of fatty acids towards synthesis of diverse metabolites across organisms.

## Results and discussion
### The promiscuous canonical CoA-binding pocket is inaccessible and redundant in FAALs

The structural features of a canonical CoA-binding pocket that allow the proper recognition of CoA/4'-PPant in different ANL superfamily members were compared and contrasted with the analogous structural positions in FAALs. A comprehensive analysis of the canonical CoA-binding pocket in the 26 structures (59 protomers) of the CoA/4'-PPant-bound ANL superfamily members (*Supplementary file 1*) revealed important aspects of CoA/4'-PPant recognition. It is observed that ligand interacts with the protein through three categories of interactions but none of the structures show the CoA/4'-PPant ligands bound in identical positions or orientations (*Figure 1—figure supplement 2a and b*). The three categories of interactions are (i) the hydrogen bonds mediated by the A8-motif of the C-terminal domain, (ii) the water-mediated contacts through the N-terminal helices H10 and H14 (notation based on *Escherichia coli* FAAL, abbreviated as *Ec*FAAL; PDB: 3PBK) (*Zhang et al., 2011*), and (iii) the interactions with the phosphates (*Gulick et al., 2003*; *Reger et al., 2007*; *Reger et al., 2008*; *Kochan et al., 2009*; *Mitchell et al., 2012*; *Li and Nair, 2015*; *Miller et al., 2016*; *Hughes and Keatinge-Clay, 2011*) of the adenosine 3',5'-bisphosphate moiety through the positively charged residues (Arg/Lys) (*Figure 1—figure supplement 3*) in the loop connecting H23 and β26 (notation based on *Ec*FAAL).

A comparative analysis of the structurally analogous CoA-binding pocket of FAALs (11 structures constituting 23 protomers) (*Arora, 2009*; *Goyal et al., 2012*; *Zhang et al., 2011*; *Li et al., 2015*; *Guillet, 2016*) sheds light on why FAALs cannot accept CoA. The analysis reveals an absence of selection from the positively charged residues (Arg/Lys) known to assist in CoA binding in the other

members of the ANL superfamily (*Figure 1a*, *Figure 1—figure supplement 3*). In addition, the CoA-binding pocket of FAALs shows the presence of bulky residues F279, W224, and M231 (in *Ec*FAAL) at the base and a unique FAAL-specific helix (FSH) (T252-R258 in *Ec*FAAL) at the entrance of the pocket (*Figure 1b*). These elements are highly conserved in other FAALs (*Figure 1c*) and restrict the space available for the incoming 4'-PPant arm of both CoA and *holo*-ACP. The superposition of CoA- and 4'-PPant-bound structures with FAALs revealed that the overcrowded pocket along with the FSH at the entrance of the pocket is unlikely to accommodate either CoA or 4'-PPant of *holo*-ACP by hindering their entry (*Figure 1b*). There are at least seven atoms of the N-terminal domain of FAALs at a distance less than 2.5 Å from the atoms of CoA in its various conformations found in the different CoA-bound FACL structures. On the contrary, the N-terminal domain of FACLs shows only one atom at 2.5 Å from the multiple CoA conformations (*Supplementary file 2*). The distance-based assessment clearly indicates the potential clashes an incoming CoA would face in the canonical pocket of FAALs. Moreover, the absence of positively charged residues will naturally be unfavorable for CoA from being appropriately oriented in the pocket. These observations lead to the hypothesis that the canonical CoA-binding pocket is rendered non-functional because of the inaccessibility in FAALs.

## Resurrecting the canonical CoA pocket enables gain of function in FAALs

The structural analysis was used to design mutations in FAALs and FACLs where the bulkier residues of the canonical CoA-binding pocket of FAALs were mutated to smaller residues to induce 'gain of function' (*Figure 2a and b*). Likewise, the smaller residues in the canonical CoA-binding pocket of FACLs were substituted with the corresponding conserved bulkier residues present in FAALs to measure the 'loss of function' (*Figure 2d and e*). Individual mutations reducing the size of residues in the canonical CoA-binding pocket resulted in the production of acyl-CoA or 'gain of function' in FAALs that otherwise does not make any acyl-CoA (*Figure 2c*). A considerable amount of the total acyl-AMP formed was converted to acyl-CoA, ~80% in the case of $MsFAAL32_{\Delta254-257}$ and ~60% in the case of $RsFAAL_{\Delta240-243}$, when the FSH segment was deleted as compared to their wild-type proteins, respectively. Some of the single-point mutants such as A253F in *Mt*FACL13 show reduced production of acyl-CoA by ~80%, while mutations in combinations such as A276F/A232M of *Af*FACL reduce the turnover of acyl-AMP to acyl-CoA by 98% as compared to wild-type (100%) (*Figure 2f*). These observations clearly indicate that the size of residues in the canonical CoA-binding pocket dictates the ability to accommodate CoA and hence the ability to facilitate the thioesterification reaction with CoA.

It was previously identified that deletion of FAAL-specific insertion (ΔFSI) can lead to gain of function in FAALs (*Arora, 2009*). Current results indicate that mutations in the canonical pocket alone are sufficient to introduce CoA production ability in FAALs. A comparison of acyl-CoA production of the ΔFSI with FSH deletion mutant (ΔFSH) (*Figure 2g*) reveals that $RsFAAL_{\Delta FSH}$ produces 2-fold excess acyl-CoA as compared to $RsFAAL_{\Delta FSI}$ while $MsFAAL32_{\Delta FSH}$ produces 10-fold excess acyl-CoA as compared to $MsFAAL32_{\Delta FSI}$ (*Figure 2h*). Therefore, it can be concluded that the FSH and other CoA-rejection elements around the canonical CoA-binding pocket significantly deter acyl-CoA production, which supersedes the inhibition of acyl-CoA production by FSI in FAALs. Overall, these observations lead to the conclusion that the available space in the canonical CoA-binding pocket of FAALs is very small to accommodate even the 4'-PPant moiety. Mutations that increase the pocket space enhance the acyl-CoA production in FAALs, while the opposite is observed in the case of FACLs. Therefore, the bulky residues in the putative canonical CoA-binding pocket of FAALs itself act as a negative selection gate to sterically exclude CoA. The rejection mechanism operational at the canonical CoA-binding pocket in FAALs is relatively more effective in rejecting CoA than the FSI-based rejection.

## Identification of an alternative 4'-PPant-binding pocket in FAALs

The steric occlusion of the 4'-PPant binding in the canonical pocket would not only result in prevention of CoA-binding but also restrict the access to the 4'-PPant tethered *holo*-ACP. Previous studies have highlighted that FAALs work in coordination with *holo*-ACP's, stand-alone or fused to polyketide synthase (PKS) or nonribosomal peptide synthetase (NRPS) modules, to produce various bioactive molecules such as complex lipids of *Mycobacterium* (*Gavalda, 2009*; *Leger, 2009*; *Simeone, 2010*; *Trivedi, 2005*; *Vats, 2012*), lipopeptides of *Ralstonia* (*Spraker et al., 2016*; *Kreutzer, 2011*; *Kage*

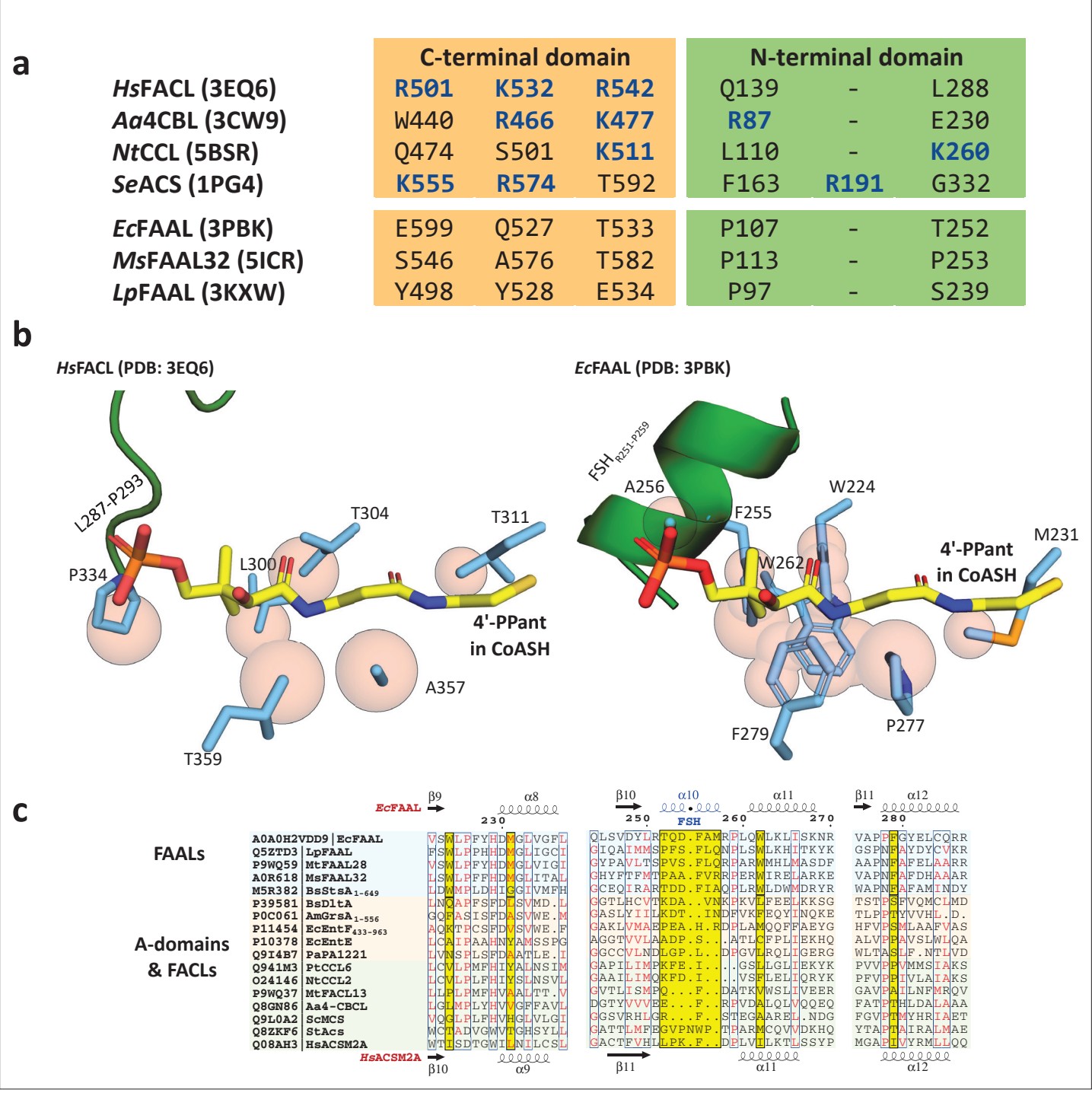

**Figure 1.** The presence of conserved negative selection elements and absence of positive selection elements in canonical coenzyme A (CoA)-binding pocket of fatty acyl-AMP ligases (FAALs) can prevent CoA binding. (**a**) The CoA-interacting residues in fatty acyl/aryl-CoA ligases (FACLs) (*Hs*FACL, *Aa*4CBL, *Nt*CCL, and *Se*ACS) and the structurally analogous residues in FAALs (*Ec*FAAL, *Ms*FAAL32, and *Lp*FAAL) are tabulated. The positively charged residues (blue) mainly are from the C-terminal domain (orange) and occasionally from the N-terminal domain (green). (**b**) The residues (cyan) in the vicinity (≤2.5 Å) of the bound CoA (yellow; *Hs*FACL; PDB: 3EQ6) are shown as van der Waals spheres (light red) to compare the canonical pocket in FAALs and FACLs. The adenosine 3',5'-bisphosphate moiety of CoA is omitted in the representation for clarity. A unique FAAL-specific helix (FSH$_{R251-P259}$) at the entry of the canonical pocket shown in cartoon representation (green) is replaced by a loop (L287-P293) in FACLs. (**c**) The canonical pocket-obstructing features seen in FAALs (light cyan) are highlighted in yellow in the structure-based sequence alignment and compared to other representative members of the ANL superfamily (FACLs in light green and A-domains in light orange). The secondary structures of *Ec*FAAL (PDB: 3PBK)

*Figure 1 continued on next page*

*Figure 1 continued*

and *Hs*FACL (PDB: 3EQ6) are marked at the top and bottom of the alignment, respectively.

The online version of this article includes the following figure supplement(s) for figure 1:

**Figure supplement 1.** A schematic showing the structural and biochemical differences between fatty acyl-AMP ligases (FAALs) and fatty acyl/aryl-CoA ligases (FACLs).

**Figure supplement 2.** Structural anlaysis of the interactions between FACLs and the bound coenzyme A shows high degree of plasticity.

**Figure supplement 3.** A surface represenation of the C-terminal domains coloured based on electrostaic potential, from FACLs and FAALs.

*et al., 2013*), and other cyanobacterial species (*Mares et al., 2014*; *Zhu et al., 2018*; *Edwards, 2004*; *Kleigrewe, 2015*; *Humbert, 2013*). Since the canonical pocket is rendered inaccessible, it immediately prompts the question, 'How then FAALs are able to transfer the activated fatty acids to the 4'-PPant arm of the *holo*-ACP?' We, therefore, sought to identify if FAALs have evolved an alternative mechanism to accommodate the 4'-PPant arm to allow binding and subsequent catalysis. Analysis of the crystal structures of the N-terminal domains of FAALs using various pocket search algorithms such as MOLE 2.0 (*Pravda, 2018*), DOGSiteScorer (*Volkamer et al., 2012*; *Fahrrolfes, 2017*), PyVOL (*Smith et al., 2019*), KVFinder (*Oliveira, 2014*), and POCASA (*Yu et al., 2010*) helped us in identifying a novel cavity in the N-terminal domain of FAALs (*Figure 3a*) but not identifiable in any of the known crystal structures of FACLs.

The entrance of the distinct tunnel is on the N-terminal side of the FSH, while the canonical pocket to accommodate CoA is on the C-terminal side of the FSH. The approach towards the active site in both cases is not on the same plane, but they coincide near the active site near the β-alanine of the 4'-PPant. The longest length along this pocket is aligned at ~25° to the canonical CoA-binding pocket. The canonical pocket is mainly formed after the rotation of the C-terminal domain in the thioesterification state (T-state) bringing the A8-motif near the active site. The space generated between the A8-motif (from the C-terminal domain) and the subdomain-B of the N-terminal domain constitutes the canonical CoA-binding pocket. In contrast, the newly identified pocket is the space between the loops in subdomain-A and the FSH region of subdomain-B of FAALs. These loop regions are highly variable in length but rich in prolines (occasionally threonine or serine) some of which are conserved in FAALs such as P226 and P107 in *Ec*FAAL (*Figure 3b*), while the FSH has a unique secondary structure characteristic of FAALs. The structurally analogous sites in FACLs show a high degree of variability with occasional prolines, but the frequency of prolines at the indicated positions is poor, as compared to FAALs, and often replaced by Asn/Leu (*Figure 3—figure supplement 1*). The ability to consistently identify the tunnel with regions enriched in prolines around the opening led to the proposition that this alternative pocket is perhaps a universal attribute of FAALs that has evolved to accommodate the 4'-PPant arm.

## The alternative pocket in FAALs is functional and accepts a 4'-PPant-tethered ACP

The ability of the alternative pocket to accommodate 4'-PPant arm tethered to ACP was tested using structure-guided mutagenesis. Bulkier residues (Phe/Arg) were introduced at the entrance of the tunnel that could potentially block the accessibility of the pocket for the incoming 4'-PPant arm. The biochemical analysis of these mutants requires an assay system to monitor the transfer of acyl-AMP to the 4'-PPant arm of *holo*-ACP. Typically, such acyl-transfers are assessed using SDS-PAGE (*Trivedi, 2004*; *Kuhn, 2016*) or conformationally sensitive urea-PAGE (CS-PAGE; *Post-Beittenmiller et al., 1991*). The ACPs that accept the acyl-chain from FAALs used in this study presented multiple complications such as poor conversion from *apo*-ACP to *holo*-ACP (*Figure 4—figure supplement 1a*) and lack of separation on a CS-PAGE (*Figure 4—figure supplement 1b*). Therefore, the CS-PAGE assay was modified for enhanced detection of the FAAL-dependent acyl-transfer on *holo*-ACP for the first time using the radiolabeled fatty acids. We tested the efficacy of the modified radio-CS-PAGE assay to probe three pairs of FAAL-ACP systems from diverse organisms (*Figure 4a*), viz., FAAL-ACP pairs from *E. coli* (*Ec*FAAL-*Ec*ACP), *Myxococcus xanthus* (*Mx*FAAL-*Mx*ACP), and *Ralstonia solanacearum* (*Rs*FAAL-*Rs*ACP). The appearance of bright bands on the radio-CS-PAGE indicates that the radiolabeled fatty acid tethered to ACP, which is absent when *apo*-ACP is used, or ATP is omitted in the

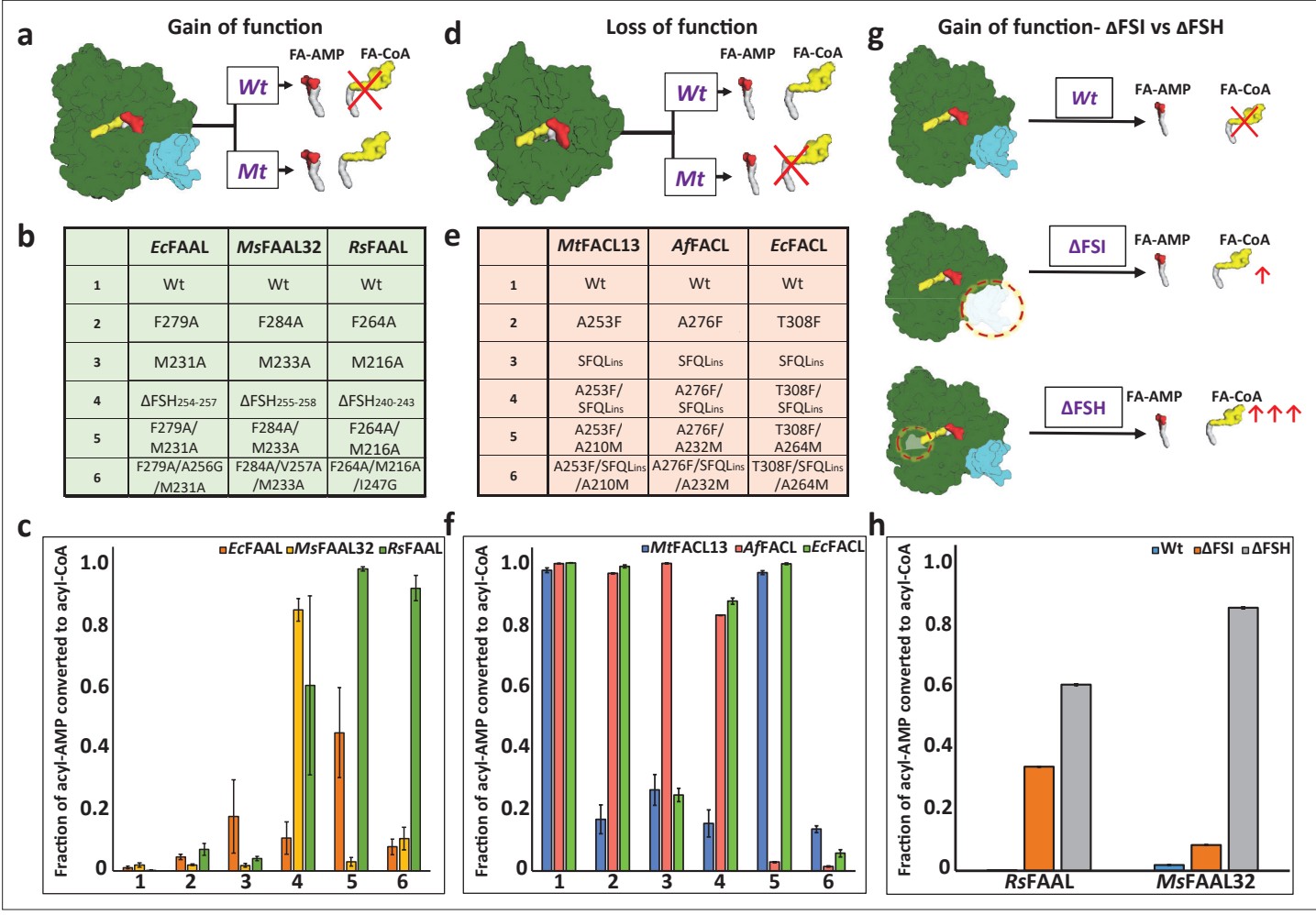

**Figure 2.** Biochemical analysis of 'gain of function' mutants of fatty acyl-AMP ligases (FAALs) and 'loss of function' mutants of fatty acyl/aryl-CoA ligases (FACLs). (**a**) The biochemical activity of a wild-type (Wt) FAAL as compared to its mutants (Mt) is schematically represented. (**b**) Various 'gain of function' mutations of FAALs (*Ec*FAAL, *Ms*FAAL32, and *Rs*FAAL) are tabulated as the following: row 1: wild -type protein; rows 2 and 3: single-point mutations; row 4: deletion of FAAL-specific helix (FSH); rows 5 and 6: combinations of these mutations. (**c**) The fraction of acyl-AMP converted to acyl-CoA by wild-type and various mutants (numbered on the x-axis according to **b**) of FAALs is represented as a bar graph along with standard error of the mean. (**d**) The biochemical activity of a wild-type (Wt) FACL as compared to its mutants (Mt) is schematically represented. (**e**) The various 'loss of function' mutations generated in FACLs (*Mt*FACL, *Af*FACL, and *Ec*FACL) are tabulated as the following: row 1: wild -type protein; row 2: single-point mutation; row 3: insertion of FSH; rows 4–6: combinations of these mutations. (**f**) The fraction of acyl-AMP converted to acyl-CoA by wild-type and various mutants (numbered on the x-axis according to **e**) of FACLs is represented as a bar graph along with standard error of the mean. (**g**) A comparison of the gain of function from ΔFSH mutation with ΔFSI mutation in FAALs is schematically represented. (**h**) The fraction of acyl-AMP converted to acyl-CoA by wild-type, ΔFSH, and ΔFSI mutants of *Rs*FAAL and *Ms*FAAL32 (represented on the x-axis) is represented as a bar graph along with standard error of the mean. The original uncropped images of these experiments are presented as *Figure 2—source data 1*. The intensity values used to compute the bar graphs are provided as an Excel file as the following: **c** = *Figure 2—source data 2* **f** = *Figure 2—source data 3*; **h** = *Figure 2—source data 4*.

The online version of this article includes the following figure supplement(s) for figure 2:

**Source data 1.** The original uncropped radio-TLC images presented here were used to assess the 'gain of function' and 'loss of function' of fatty acyl-AMP ligases (FAALs) and fatty acyl/aryl-CoA ligases (FACLs) respectively following mutations in their canonical coenzyme A (CoA)-binding pocket.

**Source data 2.** The excel file is a tabulation of the raw values of the acyl-AMP and acyl-CoA formed in the gain-of-function experiments in *Ec*FAAL, *Ms*FAAL32, and *Rs*FAAL, which are then converted to fraction of the total acyl-AMP (TA) formed to the remaining acyl-AMP (A) and the acyl-AMP converted to acyl-CoA (T).

**Source data 3.** The excel file is a tabulation of the raw values of the acyl-AMP and acyl-CoA formed in the loss-of-function experiments in *Mt*FACL13, *Af*FACL, and *Ec*FACL, which are then converted to fraction of the total acyl-AMP (TA) formed to the remaining acyl-AMP (A) and the acyl-AMP converted to acyl-CoA (T).

**Source data 4.** The excel file is a tabulation of the raw values of the acyl-AMP and acyl-CoA formed in the experiment comparing the gain of function in

*Figure 2 continued on next page*

Figure 2 continued

fatty acyl-AMP ligases (FAALs) with deletion of FAAL-specific insertion (ΔFSI) to deletion of FAAL-specific helix (ΔFSH).

**Source data 5.** The original uncropped radio-TLC showing 'gain of function' and 'loss of function' of FAALs and FACLs.

reaction. Thus, the assay system can enable the simultaneous probing of multiple FAAL-ACP pairs along with their mutations and facilitate similar studies in the future.

The modified radio-CS-PAGE was then used to test if mutating prolines (occasionally a threonine) guarding the entrance of the pocket to bulkier residues can cause abrogation of the acyl-transfer activity (*Figure 4c*). It was found that even a single-point mutation, T83F or T83R, T252F or T252R, and P107F or P107R in *Ec*FAAL, can almost abrogate the acyl-transfer reaction on *holo-Ec*ACP (*Figure 4c*). It should be noted that these mutations did not affect the adenylation ability of the proteins (*Figure 4d*). The mutations in other FAAL-ACP pairs, *Mx*FAAL-*Mx*ACP (*Figure 4—figure*

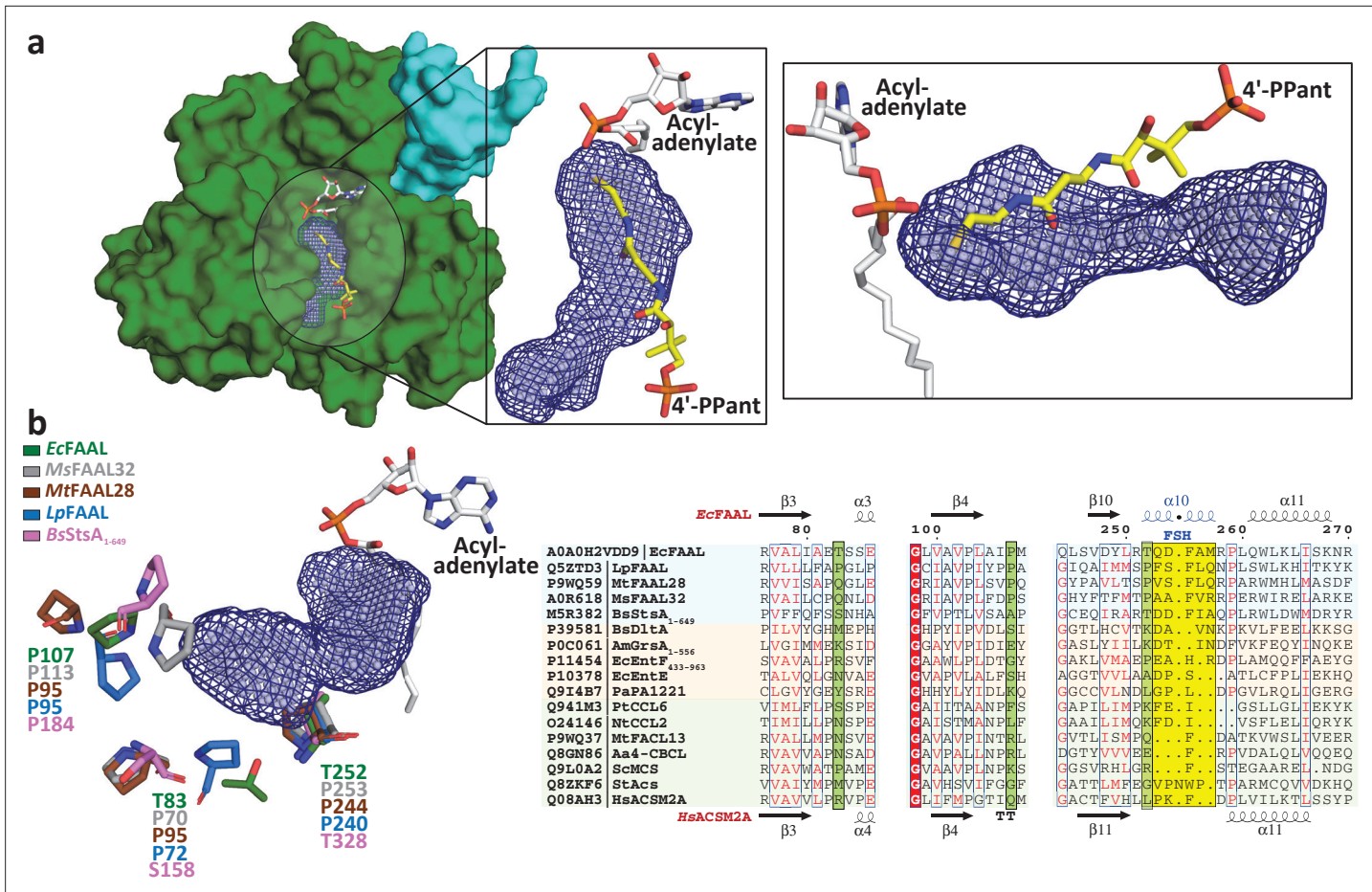

**Figure 3.** Fatty acyl-AMP ligases (FAALs) have an alternative pocket in the N-terminal domain distinct from the canonical coenzyme A (CoA)-binding pocket and lined by conserved prolines. (**a**) The analysis of N-terminal domain of *Ec*FAAL (PDB: 3PBK) (green: surface representation) with pocket finding algorithms identified a new pocket (dark blue: mesh representation; light blue: spheres representation). Also shown are the FAAL-specific insertion (cyan), an *Ec*FAAL-bound lauryl-adenylate (gray: sticks representation) and the 4'-PPant moiety (yellow: sticks representation) of a CoA bound to *Hs*FACL (PDB: 3EQ6) in the canonical pocket. The inset shows two different orientations (top view and lateral view) of the distinct alternative pocket poised against the acyl-adenylate and compared to the 4'-PPant of a CoA in canonical pocket. (**b**) Various residues (stick representation) of FAALs residing at the entrance of the unique pocket (dark blue: mesh representation; light blue: spheres representation) are identified through structural comparison. A structure-based sequence alignment of these residues (highlighted in pale green) with other representative members of the ANL superfamily reveals that FAALs have a higher frequency of prolines (occasionally Thr/Ser) than other members of the superfamily.

The online version of this article includes the following figure supplement(s) for figure 3:

**Figure supplement 1.** A comparison of the frequency of the prolines surrounding the newly identified alternative pocket in fatty acyl-AMP ligases (FAALs) (highlighted in blue) against the frequency of prolines in fatty acyl/aryl-CoA ligases (FACLs) (highlighted in green).

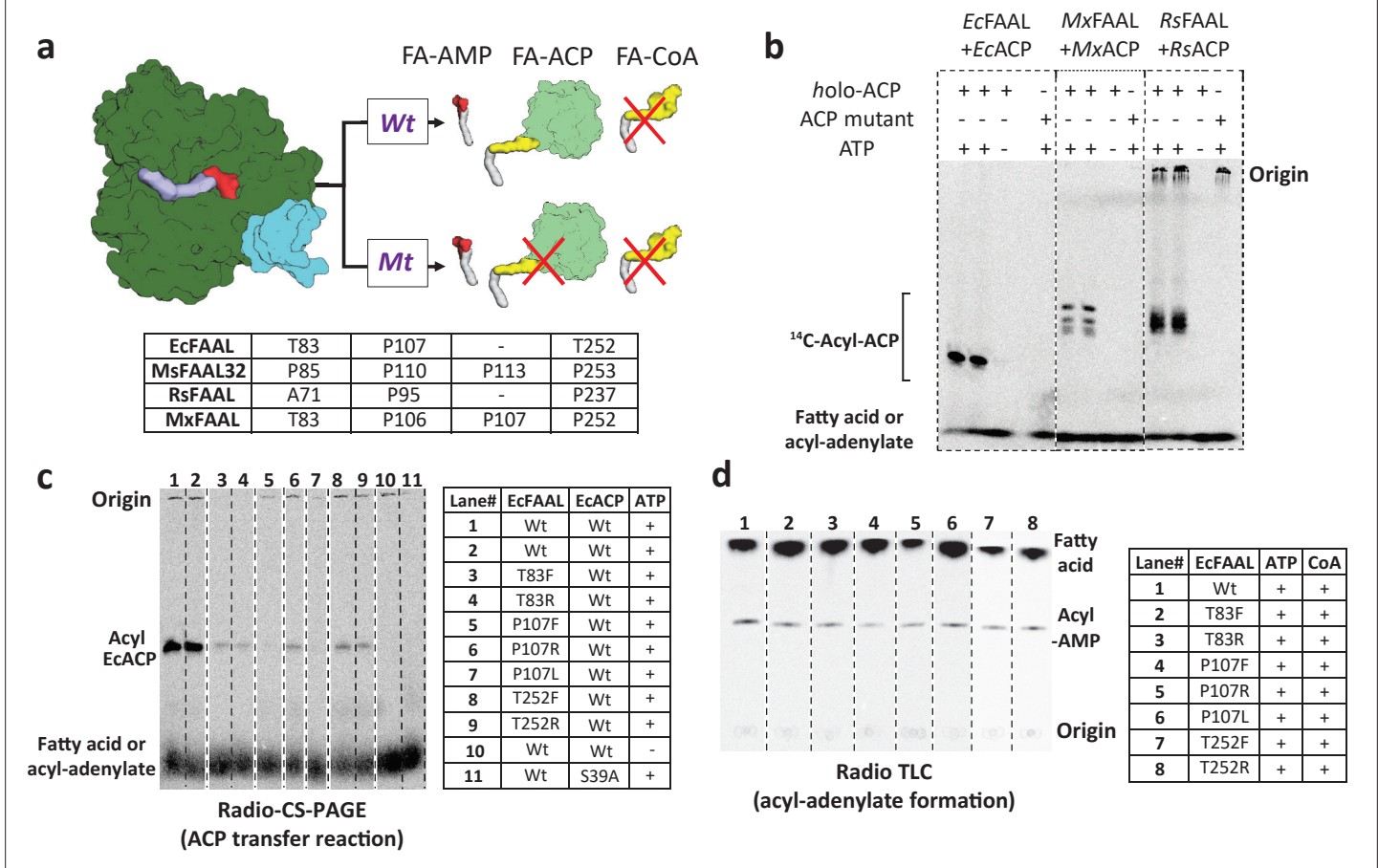

**Figure 4.** The alternative pocket identified in fatty acyl-AMP ligases (FAALs) is a functional pocket that assists in catalysis by accommodating 4'-PPant tethered to ACP. (**a**) A schematic representation of the acyl-transfer function in a wild-type (Wt) FAAL is compared to a mutant protein (Mt) that blocks the alternative pocket. (**b**) A representative radio conformationally sensitive urea-PAGE (radio-CS-PAGE) gel shows the acyl-transfer reaction by wild-type FAALs on their respective *holo*-ACP. The radio-CS-PAGE gel shows the successful transfer of the activated fatty acyl-AMP to *holo*-ACP using three pairs of FAAL-ACP systems (*Ec*FAAL-*Ec*ACP, *Mx*FAAL-*Mx*ACP, and *Rs*FAAL-*Rs*ACP), which is observed as a band due to the incorporated radiolabeled fatty acid. The absence of a band is indicative of failure to transfer as seen in the control reactions lacking ATP or the presence of a mutant ACP (conserved serine mutated to alanine) lacking the 4'-PPant arm. Samples are loaded at the point marked origin and the bands below contain either unreacted radiolabeled fatty acid or fatty acyl-AMP. (**c**) The radio-CS-PAGE shows that mutations of the residues lining the alternative pocket to Arg/Phe show a reduced or negligible acyl-ACP formation as the bands are diminished or absent. (**d**) A radio-TLC shows the acyl-AMP formation ability of various FAAL mutants indicated by the bright band in the center after the sample was loaded at the origin and developed until the edge of TLC where unused radiolabeled fatty acid is observed. The radio-TLC shows that loss of acyl-ACP formation is not due to affected acyl-adenylate formation. The original uncropped images of these experiments are presented as *Figure 4—source data 1*.

The online version of this article includes the following figure supplement(s) for figure 4:

**Source data 1.** The radio conformationally sensitive urea-PAGE (radio-CS-PAGE) images presented here were used to show the optimization of fatty acyl-AMP ligase (FAAL)-dependent acyl transfer on *holo*-ACP using three different FAAL-ACP systems (*Ec*FAAL-*Ec*ACP, *Rs*FAAL-*Rs*ACP, and *Mx*FAAL-*Mx*ACP).

**Figure supplement 1.** Typically, acyl-transfer reactions with ACP are assessed using an SDS-PAGE (coupled to radio-labeled or fluorescently labeled substrates) or conformationally sensitive urea-PAGE or HPLC-coupled to mass spectrometry.

**Figure supplement 1—source data 1.** A representative image of Coomassie-stained conformationally sensitive urea-PAGE (15% acrylamide and 2.5 M urea) of the *Ec*FAAL catalyzed acyl-transfer on holo-*Ec*ACP.

**Figure supplement 2.** The radio-TLC images presented here were used to assess the ability of fatty acyl-AMP ligases (FAALs), *Mx*FAAL (**A**), *Rs*FAAL (**B**), and *Ms*FAAL32 (**C**), with mutations in the alternate pocket to form acyl-AMP.

**Figure supplement 2—source data 1.** The FAAL-ACP pairs, *Rs*FAAL-*Rs*ACP and *Mx*FAAL-*Mx*ACP, along with mutations in the alternate pocket of the respective fatty acyl-AMP ligases (FAALs), were assessed using modified radio conformationally sensitive urea-PAGE (CS-PAGE) while *Ms*FAAL32-*Ms*PKS$_{1-1042}$ was assessed using a radio-SDS-PAGE.

*supplement 2a and b*) and *Rs*FAAL-*Rs*ACP (*Figure 4—figure supplement 2c and d*), also resulted in similar abrogation of the acyl-transfer ability on their respective cognate *holo*-ACPs. An additional FAAL-PKS pair of *Ms*FAAL32-*Ms*PKS13$_{1-1042}$ was also mutated and probed biochemically using the traditional radio-SDS-PAGE (*Figure 4—figure supplement 2e and f*). The mutations of residues guarding the alternative pocket to bulkier residues in the *Ms*FAAL32-*Ms*PKS13$_{1-1042}$ system also resulted in diminished acyl-transfer ability. These results indicate that blocking the entrance of the alternative pocket prevents the entry of the incoming 4'-PPant arm tethered to ACP. Therefore, in FAALs, the identified alternative pocket is fully functional and distinct from the non-functional canonical pocket. Such a unique pocket facilitates the entry of the 4'-PPant arm of the ACP to approach the active site and catalyze the acyl-transfer reaction, a feature absent in other members of the superfamily.

## A universal mechanism for rejection of highly abundant CoA in the alternative pocket

The primary attribute of a functional alternative pocket in FAALs is to discriminate and reject CoA from the chemically identical 4'-PPant of *holo*-ACP. Therefore, the pocket should allow the entry of 4'-PPant into the tunnel but not the additional 'head group,' adenosine 3',5'-bisphosphate moiety, containing CoA. Structural analysis of eight CoA-bound protein structures (constituting 14 protomers) of the ANL superfamily reveals the variability or degrees of freedom available for adenosine 3',5'-bisphosphate moiety (head group). The analysis reveals that the 4'-PPant arm remains largely rigid in an extended form and the major source of variability is the conformational freedom around the 4'-phosphate (*Figure 5—figure supplement 1a*). Such a variability has previously been used to classify the conformations of CoA as extended conformations (e.g., *Se*ACS, *Hs*FACL) or bent conformations (e.g., *Aa*4'-CBL) (*Chen, 2017*). Limited variabilities in the conformations of the adenine ring and ribose are clustered in a small zone and therefore can be ignored (*Figure 5—figure supplement 1a*). The length of the 4'-PPant arm is around 18–20 Å (*Leibundgut et al., 2007*) while it is around 15–16 Å (*Mitchell et al., 2012*) as seen in the crystal structures of various ANL superfamily members, which is comparable to the length of predicted pockets (17–18 Å). The biochemical evidence (*Figure 4c*) along with length considerations indicates that the 4'-PPant arm can be accommodated within the predicted pocket. In the absence of structural information, the orientation of the 4'-PPant arm is random but limited to space with the identified pocket such that the 4'-phosphate is at the entry of the tunnel and the thiol near the active site.

Based on the above considerations, a CoA molecule in ANL superfamily members can be visualized as 'flag hoisted on a mast.' The extended 4'-PPant arm can be considered as the 'mast' and the adenosine 3',5'-bisphosphate moiety as the 'flag.' The rotation of the 'flag' around the 'mast' placed within the pocket is comparable to the degree of freedom available for the 'flag' (*Figure 5—figure supplement 1b*). We, therefore, rotated the 'flag' around the 'mast' to generate conformations at an angular sampling rate of 1° per conformation for each of the four known conformations of CoA (*Figure 5a*). It was found that main-chain atoms and Cβ atoms of the FAAL protein, irrespective of the conformation of the C-terminal domain (A-state or T-state), show an average of 28 clashes (van der Waals overlap >0.25 Å) (*Figure 5b*, *Figure 5—figure supplement 2a*). The adenosine 3',5'-bisphosphate or the 'flag' would encounter severe clashes because the entry of the alternative pocket is deeply embedded within the subdomain-A and -B of the N-terminal domain. It is unlike the entry of the canonical pocket, which is entirely open with the subdomain-B forming the base of the pocket (*Figure 4*). Therefore, the N-terminal domain of FAALs itself forms a strong deterrent for accommodating adenosine 3',5'-bisphosphate and hence prevents CoA binding (*Figure 5—figure supplement 2b*). Interestingly, the structural elements resisting the adenosine 3',5'-bisphosphate are unique to FAALs such as the FSH (T252-R258 in *Ec*FAAL) and the loop harboring proline residues that guard the opening of the tunnel. Therefore, a strong negative selection and the absence of any positive selection make accommodation of adenosine 3',5'-bisphosphate containing CoA unlikely. In contrast, the adenosine 3',5'-bisphosphate lacking 4'-PPant of *holo*-ACP can easily access the pocket and participate in the acyl-transfer reaction. This essentially forms the structural basis for the CoA-rejection mechanism in FAALs (*Figure 5c*).

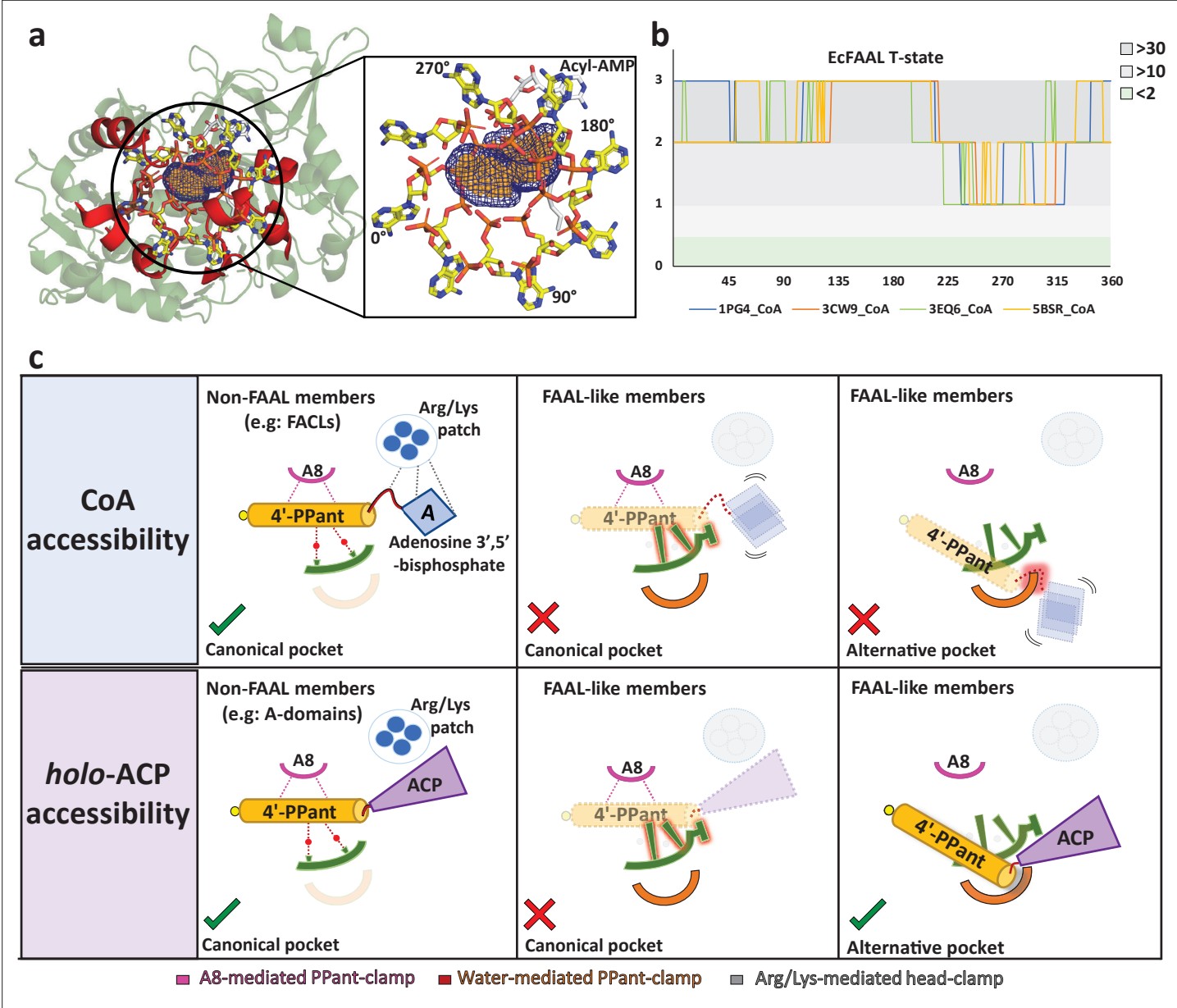

**Figure 5.** The alternative pocket in fatty acyl-AMP ligases (FAALs) is highly selective, and its unique architecture negatively selects for coenzyme A (CoA). (**a**) A 'mast' (the 4'-PPant arm) is aligned along the longest length of the predicted pocket (blue: mesh representation) and the 'flag' (adenosine 3',5'-bisphosphate) is rotated to generate theoretical orientations of the 'flag.' These conformations of *Hs*FACL (PDB: 3EQ6) are shown (yellow: stick format) at 45° intervals. The regions of the *Ec*FAAL (cartoon representation) showing clashes are highlighted (red). (**b**) The count of atoms (C, N, O, Cα, and Cβ) of *Ec*FAAL showing van der Waals short contacts (≤0.25 Å) with the conformations of the 'flag' is scored (short contacts ≥30 = 3; ≥10 = 2; ≥2 = 1 and <2 = 0). These scores (y-axis) are plotted against the specific rotation angles of the conformation, where the short contact is observed (x-axis). (**c**) The universal CoA-rejection mechanism is schematically summarized. The 4'-PPant (light orange) of both CoA and *holo*-ACP (purple) bind to the canonical pocket of non-FAAL members through multiple interactions (dotted lines). Interactions between Arg/Lys patch (circle: blue) and the adenosine 3',5'-bisphosphate moiety (rhombus: blue) acts as positive selection. The negative selection elements of the canonical pocket clash (highlighted in red) with the adenosine 3',5'-bisphosphate in the alternative pocket. The absence of Arg/Lys patch (dotted circle) fails to provide stability to the adenosine 3',5'-bisphosphate moiety (dotted rhombus; blue). The 4'-PPant (yellow) tethered to ACP (purple) is only accepted in the alternative pocket, which is absent in non-FAAL members, as it shows no clashes. The ranked count of atoms used to generate the plot **5b** is provided as an Excel file, *Figure 5—source data 1*.

The online version of this article includes the following figure supplement(s) for figure 5:

**Source data 1.** The Excel sheet tabulates the number of clashes observed between various conformations of coenzyme A (CoA) and fatty acyl-AMP ligases (FAALs) (main-chain atoms: N, C, O, Cα, and Cβ).

*Figure 5 continued on next page*

Figure 5 continued

**Figure supplement 1.** Structural analysis of known coenzyme A conformations and a schematic showing accomodation of 4'-phosphopantetheine arm in the alterative pocket.

**Figure supplement 2.** The clash score analysis of various conformations of adenosine 3',5'-bisphosphate shows the incompatibility of coenzyme A binding in alternative pocket.

## Identification of FAAL-like proteins across different forms of life

Previous studies show that FAALs can be identified based on the presence of insertion and its anchorage to the N-terminal domain through hydrophobic interactions. A lack of sequence conservation in the FSI and only a single structural template for further analysis complicated the search procedure. Therefore, early genome mining efforts resulted in a sparse identification of FAALs in lineages of actinobacteria, cyanobacteria, and proteobacteria along with some eukaryotes such as humans and mouse (*Goyal et al., 2012*). The conserved sequence features from this study such as FSH, a blocked canonical pocket, and a proline-lined 4'-PPant binding pocket led to the identification of FAAL-like proteins with greater confidence. The analysis revealed a ubiquitous distribution of FAALs across different forms of life including bacteria, plants, fungi, and animals, except archaea (*Figure 6a*). The phylogenetic analysis reveals that all the FAAL-like proteins diverge and cluster away from FACLs or A-domains. FAAL-like proteins show three subgroups, viz., bacterial/plant FAALs group, fungal FAAL-like protein group, and animal FAAL-like protein group (*Figure 6a*). Typical FAALs are bacterial FAALs, which are always found in a genomic context with PKS and PKS/NRPS hybrid genes either as a stand-alone domain or as a didomain fused to ACP or multidomain fused to the entire PKS/NRPS gene (*Supplementary file 3*). Occasionally, the stand-alone bacterial FAALs are found to be interspersed with additional domains (*Figure 6—figure supplement 1a*), mainly with dehydrogenases and oxygenases. Most plant FAAL-like domains resemble the bacterial FAALs from their sequence identity and domain organization. However, unique domain organizations are also found in plant FAAL-like domains, where an uncharacterized protein has FAAL-like domains sandwiched between HemY domains and catalase domains or amino acid oxidase domains (*Figure 6—figure supplement 1a*). To the best of our knowledge, this is the first report of the presence of FAAL-like domains in plants and also the fusion of alternate domains to FAALs at their N-terminus.

Fungal and animal FAAL-like domains tend to cluster together with bacterial/plant FAAL-like domains in bioinformatic analysis (*Figure 6a*, *Figure 6—figure supplement 1b*) because of the conservation of unique FAAL-specific features. Despite their lower sequence identity, the FSI (*Figure 6—figure supplement 1c and d*) supported by a hydrophobic patch and the FSH and CoA-rejecting features (*Figure 6—figure supplement 2a and b*) are clearly identifiable. Such conserved FAAL-specific features indicate that FAAL-like modules are recruited in eukaryotes for specific metabolic processes owing to their unique ability to reject CoA. Interestingly, fungal and animal FAAL-like domains share a highly conserved three-domain architecture, and such an architecture is not seen in any of the prokaryotes and plants (*Supplementary file 4*). The conserved three-domain protein consists of a N-terminal DMAP1-binding domain followed by a tandemly fused two FAAL-like domains as found in fungi to animals. The eukaryotic metabolic context for avoiding a reaction with CoA through these FAAL-like domains and their unique domain organization remains to be explored. Based on these observations, FAAL-like domains are more widespread than previously anticipated and their omnipresence is comparable to the universally present FACLs. Therefore, we propose that FAALs may not have descended from FACLs and they rather share a parallel evolutionary history. The ancestral ANL fold could have diverged as non-promiscuous FAAL-like members and promiscuous nonFAAL-like members simultaneously in the last universal common ancestor (*Figure 6b*).

## Conclusions

The study uncovers the mechanistic basis of how FAALs strictly reject CoA, which is highly abundant and almost chemically identical to their actual substrate, *holo*-ACP. The rejection mechanism relies on a discriminatory 4'-PPant-accepting pocket while avoiding the promiscuous canonical CoA-binding pocket that is rendered non-functional. It has been achieved through bulky hydrophobic residues in the pocket and a unique secondary structural element, FSH, at the entrance. The discriminatory 4'-PPant-accepting pocket in FAALs, on the other hand, has a unique architecture that negatively

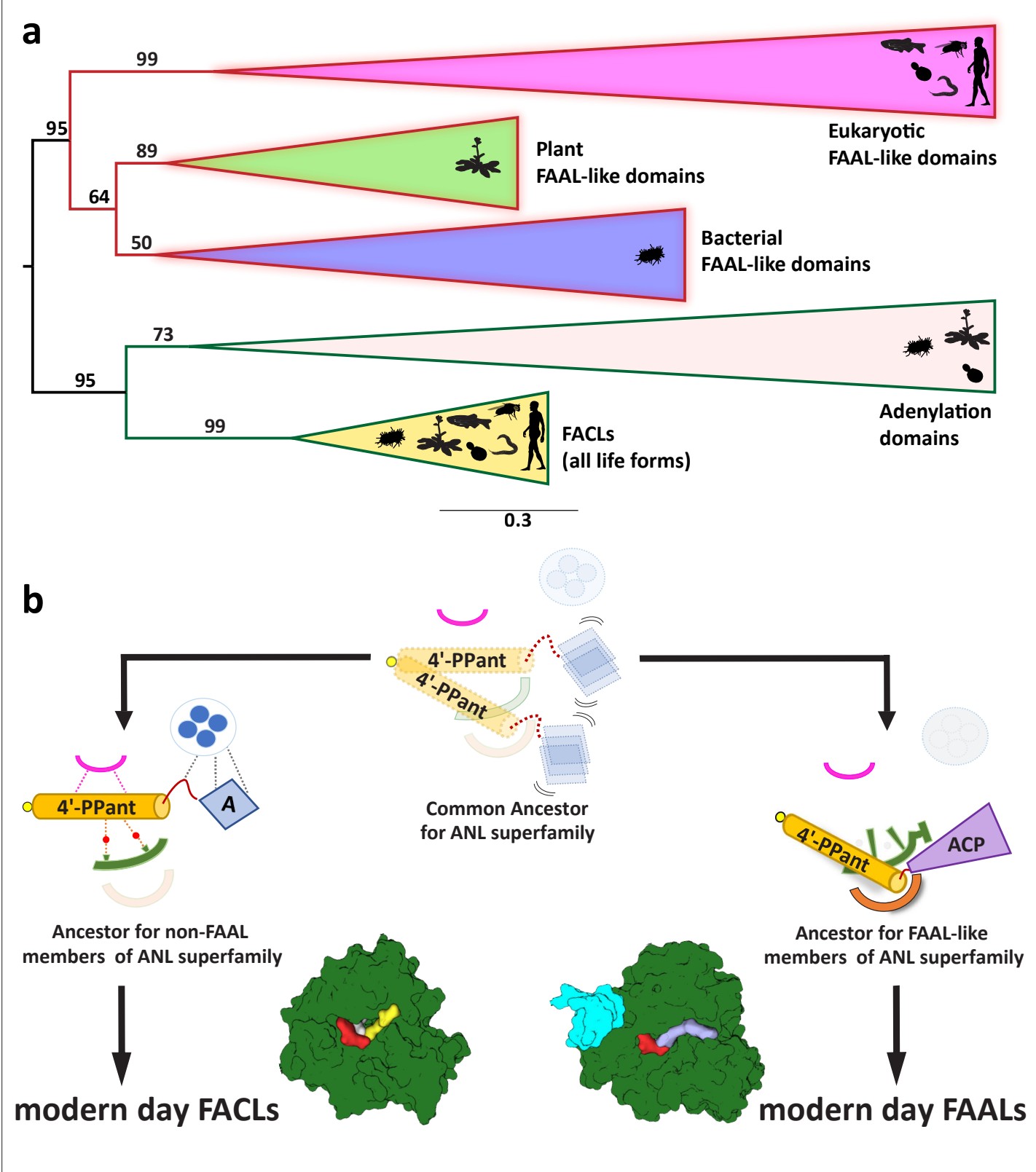

**Figure 6.** The coenzyme A (CoA)-rejection elements in fatty acyl-AMP ligases (FAALs) are conserved in all forms of life, and therefore, FAALs and fatty acyl/aryl-CoA ligases (FACLs) have parallelly evolved from a common ancestor of the ANL superfamily. (**a**) A clustering diagram with the bootstrapping values for all the FAAL-like sequences is presented. The ANL superfamily members have two major divergent classes, viz., CoA-rejecting FAAL-like sequences and CoA-accepting non-FAAL members (FACLs and A-domains). A graphical representation of the distribution of these members is

*Figure 6 continued on next page*

*Figure 6 continued*

shown using pictorial representation of the organisms, which include bacteria, plants, yeast, worm, fly, and human. (**b**) The ancestral fold of the ANL superfamily may have been a loose organization of peptide scaffolds working with thiol-containing molecules such as pantetheine, etc. The ancestral fold accumulated various mutations, resulting into the ancestor of modern-day acceptor promiscuity lacking FAAL-like forms and acceptor promiscuity containing non-FAAL members. The evolution of acceptor promiscuity spectrum may have been driven based on their participation in bulk metabolic reactions. FAALs did not participate in bulk metabolic reactions and hence dedicated themselves to their cognate ACP partners to redirect small molecules to specific pathways. The non-FAAL members of the ANL superfamily such as FACLs participate in bulk reactions, where minor cross-reaction products are observed.

The online version of this article includes the following figure supplement(s) for figure 6:

**Figure supplement 1.** The fatty acyl-AMP ligases (FAAL)-like sequences were identified and aligned to generate a structure-based alignment as described in the Materials and methods section.

**Figure supplement 2.** The fatty acyl-AMP ligases (FAAL)-like sequences were identified and aligned to generate a structure-based alignment as described in the Materials and methods section.

**Figure supplement 3.** The fatty acyl-AMP ligases (FAAL)-like sequences were identified and aligned to generate a structure-based alignment as described in the Materials and methods section.

selects adenosine 3',5'-bisphosphate moiety and also lacks Arg/Lys residues for positive selection. Interestingly, these rejection criteria are not only conserved in bacteria but also in all forms of life (excluding archaea). The unique remodeling of pockets to ensure acceptor discrimination probably allowed the evolutionary recruitment of FAAL-like domains in metabolic crossroads for redirecting the fate of molecules to specific pathways. Such a widespread conservation of FAAL-like proteins puts the evolutionary origin of FAALs parallel with FACLs in the last universal common ancestor and not as a subset of FACLs.

The scaffold of the ANL superfamily of enzymes is known to be promiscuous not only for the substrates they act on (*Arora et al., 2005*), but also the final acceptor of the acyl adenylates. Surprisingly, billions of years of evolution have not prevailed upon the substrate-promiscuity problem as well as the acceptor-promiscuity problem. The probability of acceptor promiscuity influencing the erroneous product formation is high as some of the acceptors such as CoA and pantetheine are the most abundant molecules in the cell. The problem is greatly amplified as it can redirect the fate of metabolites from one pathway to another such as primary metabolism to secondary metabolism. Our analysis shows that the CoA-bound structures of ANL superfamily members exhibit non-conformity in the binding mode of 4'-PPant or the adenosine 3',5'-bisphosphate moiety (*Reger et al., 2008*; *Zahn, 2019*). The variability in the 4'-PPant binding is also true for the A-domain:ACP complexes (*Mitchell et al., 2012*; *Sundlov et al., 2012*; *Sundlov and Gulick, 2013*). The variability could be resulting from highly divergent residues lining the CoA-binding site in FACLs and their analogous sites in FAALs (*Supplementary file 5*), which possibly explains the observed differential activity of certain loss-of-function and gain-of-function mutations (*Supplementary file 5*). Hence, it is evident that a defined CoA-binding pocket is lacking in ANL superfamily members, which is commensurate with the failure to identify the pocket in FACLs using various pocket search algorithms. Therefore, a promiscuous access to the active site without any selection determinants is the likely cause for the lack of discrimination for final acceptors in these enzymes.

It is not clear if the observed spectrum of latent activities in these enzymes' design is an evolutionary relic or has any physiological relevance in a specific cellular context. The persistence of acceptor promiscuity can only have two explanations: viz., the cross-reaction products are beneficial, or pocket modification is not possible without compromising the basic function. In this context, FAALs are surprisingly high-fidelity enzymes representing the extreme end of the promiscuity spectrum, offering no cross-reaction with CoA as acceptors of the acyl adenylates (*Trivedi, 2004*; *Arora, 2009*; *Goyal et al., 2012*; *Priyadarshan and Sankaranarayanan, 2018*). A functional and discriminatory 4'-PPant binding in FAALs has established them as an alternative enzymatic bridge between FA synthesis, exogenous fatty acids import, and PKS/NRPS machinery that produce diverse bioactive natural products such as bacillomycin (*Lu, 2019*; *Koumoutsi, 2004*), mycosubtilin (*Duitman, 1999*), daptomycin (*Wittmann et al., 2008*; *Baltz, 2021*), phenylnannolone (*Bouhired, 2014*), alkylresorcylic acid (*Hayashi et al., 2011*), caryoynencin (*Ross et al., 2014*), vioprolide (*Auerbach et al., 2018*), taromycin A (*Yamanaka, 2014*), tambjamine (*Marchetti et al., 2018*), ambruticin (*Hemmerling et al., 2018*; *Julien et al., 2006*), , to name a few (*Supplementary file 3*). The relevance of the lack of

acceptor promiscuity in FAALs has been demonstrated in *Mycobacteria* as being responsible for dictating the fate of free fatty acids in producing virulent lipids (*Arora, 2009*).

The conserved sequence and structural features of bacterial FAALs enabled better identification of FAAL-like domains across all forms of life, except archaea. FAALs may have been recruited to specifically 'load' fatty acids on PKS/NRPS but the possibility of lack of promiscuity in FAALs driving their over-representation cannot be ignored. Their characteristic absence in archaea perhaps can be explained by the abysmal frequency or absence of PKS or NRPS systems. FAAL-like domains of bacteria and plants exhibit remarkable sequence similarity, which is expected because of the presence of canonical PKS/NRPS enzymology in both systems. A few of the identified FAAL-like domains in bacteria show unique domain organizations such as a fusion with lysophospholipid acyltransferases and alpha aminoadipate reductases or found interspersed in operons with dehydrogenases, decarboxylase, and ketoacyl synthases. Unique domain architectures are also found in plants, where FAAL-like domains are sandwiched between HemY and catalase domains and the function of these conserved proteins remain unknown. These unique architectures are evolutionary instances of the recruitment of FAAL-like domains for PKS/NRPS-independent functions.

Interestingly, unique domain architectures are found in FAAL-like domain-containing proteins of fungi and animals, the majority of which are largely PKS/NRPS-free systems. In these organisms, two tandemly fused FAAL-like domains are found at the C-terminus and a DMAP-1-binding domain at the N-terminus of a highly conserved protein. These proteins are annotated and characterized as the virulent factor CPS1 in few fungal pathogens such as *Magnaporthe oryzae* (*Wang, 2016*), *Cochliobolus heterostrophus* (*Lu, 2003*), and *Coccidioides posadasii* (*Narra, 2016*). Therefore, they are proposed as a target for potential fungicides and as a vaccine candidate in plant and animal pathogens, respectively (*Wang, 2016*; *Lu, 2003*; *Narra, 2016*). In invertebrates and vertebrates, these are known as DIP2 and are important for proper axon bifurcation and guidance, suggesting their physiological importance (*Nitta et al., 2017*; *Noblett, 2019*). Recently, we showed that the mice lacking one of the eukaryotic FAAL-like proteins, DIP2A, show a diet-dependent growth anomaly such as obesity pointing to their importance in eukaryotic lipid metabolism (*Kinatukara, 2020*). The precise molecular and biochemical details of these proteins are yet to be elucidated. Given the extent of sequence divergence in various motifs of eukaryotic FAAL-like domains (*Figure 6—figure supplement 3a–d*), it is difficult to predict the processes they may be involved in. However, the presence of several elements of the CoA-rejection mechanism and their close identity to FAALs allow us to put forth the hypothesis that eukaryotic FAAL-like domains were recruited for their high acceptor-fidelity property. It is also possible that these divergences have some functional relevance in the context of eukaryotic metabolism, which may represent an additional member in the spectrum of biochemical activities represented in the ANL superfamily.

The current work identifying FAAL-like enzymology using a universal CoA rejection mechanism may form the platform for further studies to delineate why they have been recruited in fungal and animal systems. The study provides new structural and sequence attributes to confirm the identity of FAALs, many of which remain misannotated and uncharacterized. The study opens new avenues in combinatorial engineering of PKS/NRPS by using FAALs as a unique module to load fatty acids or use preexisting knowledge to engineer them to load unique molecules (*Clark et al., 2018*) with exceptional fidelity. Thus, FAALs can be exploited to produce novel bioactive molecules by virtue of their unique acceptor-fidelity property.

# Materials and methods
## Cloning, expression, and purification of proteins

*Ec*FAAL (A0A0H2VDD9), *Ec*ACP (A0A2X1NC35), *Ec*FACL (P69451), *Ms*FAAL32 (A0R618), *Ms*PKS13 (A0R617; 1–1042 residues), *Rs*FAAL (Q8XRP4), *Rs*ACP (Q8XRP0), *Af*FACL (O30147), *Mx*FAAL (Q1CXX0), *Mx*ACP (Q1CXW9), and *Mt*FACL13 (P9WQ37) were amplified by PCR from their respective genomic DNA using Phusion polymerase (Thermo Scientific) using specific primers (*Supplementary file 6*). These genes were expressed as hexahistidine-tagged proteins after the induction with IPTG using the *E. coli* BL21(DE3) expression system. The mutants were generated using quick-change site-directed mutagenesis. All the proteins including mutants were expressed and purified to homogeneity

using Ni-NTA affinity chromatography followed by size-exclusion chromatography at 4 °C. They were flash-frozen in liquid nitrogen and stored at –80 °C until further use.

## Biochemical analysis of FAALs and FACLs

The acyl-AMP and acyl-CoA formation by FAALs and FACLs was performed by previously described methods (*Trivedi, 2004*). Briefly, the reaction was carried out in a 15 µL reaction volume with 50 mM Tris (pH 8.0) containing 8 mM MgCl$_2$, 176 µM fatty acid, and 24 µM 1–14C fatty acid. Lauric acid (C12) was used for all enzymes except MsFAAL32 where palmitic acid (C16) was used as the substrate. 2 mM CoASH was either included or excluded from the reaction according to the reaction process being assessed. 2 mM ATP was used to initiate the reaction and incubated at 30 °C for 15 min on the water bath. The reaction was quenched using 5 µL of 10% acetic acid. The entire 20 µL reaction mixture was directly spotted on a silica gel 60 F$_{254}$ TLC plate and allowed to dry for 2 hr. Subsequently the products were resolved using n-butanol/acetic acid/water (80:25:40; v/v) solvent system at 4 °C. All FACLs (*Mt*FACL13, *Ec*FACL, *Af*FACL), *Ec*FAAL and *Ms*FAAL32 were used at 5 µM concentration while *Mx*FAAL and *Rs*FAAL were used at 7.5 µM. The $^{14}$C-fatty acids allowed detection of the products on a phosphorimager (Amersham Typhoon FLA 9000), which were then quantified by densitometry using Image Lab (Bio-Rad Laboratories Inc). All experiments were performed as triplicates. The percentage of acyl-AMP converted to acyl-CoA from the total acyl-AMP formed is used to plot and compare the activity, along with the standard error of the mean, of wild-type against the respective mutants.

## Conversion of *apo*-ACP to *holo*-ACP

All the purified ACPs were converted to *holo*-ACP before being flash-frozen using previously described protocols (*Lambalot, 1996*). Briefly, after Ni-NTA purification, they were buffer exchanged to the phosphopantetheinylation buffer (20 mM Tris pH 8.8, 10 mM MgCl$_2$, and 10 mM dithiothreitol). Approximately 250 µM of ACP was then incubated with 1.25 µM of a non-specific phosphopantetheinyl transferase *Sfp* from *B. subtilis* in a 300 µL reaction mixture along with 10 mM DTT, 10 mM MgCl$_2$, and 1 mM CoASH. The reaction mixture was incubated at 25 °C for 12–16 hr and further purified using size-exclusion chromatography at 4 °C. The *holo*-ACP was flash-frozen in liquid nitrogen and stored at –80 °C until further use. The conserved serine on which the 4'-PPant moiety is post-translationally modified was mutated to alanine, and the resulting protein was used as *apo*-ACP for all assays.

## Loading-activated acyl chains on *holo*-ACP by FAALs

The transfer of activated acyl-chain to stand-alone *holo*-ACP by FAALs was assessed using radiolabeled fatty acids combined with CS-PAGE (*Post-Beittenmiller et al., 1991*). The following ratio of FAAL to *holo*-ACP was used: 1 µM of *Ec*FAAL with 20 µM of *Ec*ACP, 4 µM of *Mx*FAAL with 8 µM of *Mx*ACP, 5 µM of *Rs*FAAL with 20 µM *Rs*ACP and 1 µM of *Ms*FAAL32 with 12.5 µM of *Ms*PKS13ΔC. These proteins were incubated in a typical 15 µL reaction mixture composed of 50 mM HEPES pH 7.2, 5 mM MgCl$_2$, 0.01% w/v Tween-20, 0.003% DMSO, 2 mM ATP, and 9 µM of $^{14}$C labeled-lauric acid (C12). The reactions of *Mx*FAAL-*Mx*ACP and *Rs*FAAL-*Rs*ACP consisted of 20 mM Tris pH 8.0 and lacked DMSO. The reaction mixture was incubated for 2 hr at 30 °C and quenched with an equal volume of urea loading dye (25 mM Tris pH 6.8, 1.25 M urea, 20% glycerol, and 0.08% bromophenol blue). The contents were immediately loaded and separated on a 15% PAGE containing 2.5 M urea. The acyl-transfer on *holo*-*Ms*PKS13$_{1-1042}$ by *Ms*FAAL32 was carried out as previously described (*Kuhn, 2016*). Briefly, $^{14}$C labeled-palmitic acid (C16) was used to assess the transfer on *holo*-*Ms*PKS13$_{1-1042}$ on an 8% SDS-PAGE. All gels were then dried using a gel drier (Bio-Rad gel dryer 583) and the radiolabeled *acyl*-ACP was detected using the phosphorimager (Amersham Typhoon FLA 9000).

## Sequence and structural analysis

All sequences were identified and retrieved from the NCBI sequence database using *Ec*FAAL as template and BLAST search algorithm. A structure-based sequence alignment was generated using msTALI (*Shealy and Valafar, 2012*) and then other sequences were added to the alignment in MAFFT (*Katoh et al., 2019*). The sequence alignments were rendered using ESPript (*Robert and Gouet, 2014*). The phylogenetic analysis of these sequences was carried out using both neighbor-joining as implemented in MAFFT (*Katoh et al., 2019*) and maximum-likelihood as implemented in IQTREE (*Nguyen et al., 2015*; *Hoang et al., 2018*; ) using default parameters. The phylogenetic tree was

rendered in MixtureTree Annotator (*Chen and Ogata, 2015*). A graphical representation of the multiple sequence alignment was converted to sequence logo using WebLogo (*Crooks et al., 2004*). Images of various organisms were reusable silhouette images of organisms obtained from Phylopic (https://www.phylopic.org) under Creative Commons license. All structural analyses including structural superposition, van der Waals distance measurements, and ligand alignment were carried out with various in-built features of PyMOL (*Schrödinger, LLC, 2015*). All cavity search programs were run using default parameters.

## Acknowledgements

We acknowledge Dr. Pananghat Gayathri (IISER-Pune) and Dr. Raghunand R Tirumalai (CSIR-CCMB) for sharing the genomic DNA samples of *Myxococcus xanthus* and *Mycobacterium smegmatis*, respectively.

## Additional information

### Competing interests

Rajan Sankaranarayanan: Reviewing editor, *eLife*. The other authors declare that no competing interests exist.

### Funding

| Funder | Grant reference number | Author |
|---|---|---|
| Department of Biotechnology, Ministry of Science and Technology, India | | Gajanan S Patil |
| Council of Scientific and Industrial Research, Ministry of Science and Technology, India | | Sudipta Mondal Rajan Sankaranarayanan |
| University Grants Commission | | Sakshi Shambhavi |
| Science and Engineering Research Board, India | J. C. Bose Fellowship | Rajan Sankaranarayanan |

The funders had no role in study design, data collection and interpretation, or the decision to submit the work for publication.

### Author contributions

Gajanan S Patil, Priyadarshan Kinatukara, Conceptualization, Data curation, Formal analysis, Investigation, Methodology, Validation, Visualization, Writing – original draft, Writing – review and editing; Sudipta Mondal, Data curation, Formal analysis, Investigation, Validation, Visualization, Writing – review and editing; Sakshi Shambhavi, Formal analysis, Investigation, Validation, Visualization, Writing – review and editing; Ketan D Patel, Data curation, Formal analysis, Validation, Visualization; Surabhi Pramanik, Noopur Dubey, Data curation, Investigation, Methodology, Writing – review and editing; Subhash Narasimhan, Murali Krishna Madduri, Formal analysis, Investigation, Writing – review and editing; Biswajit Pal, Rajesh S Gokhale, Formal analysis, Validation, Visualization, Writing – review and editing; Rajan Sankaranarayanan, Conceptualization, Data curation, Formal analysis, Funding acquisition, Investigation, Methodology, Project administration, Resources, Supervision, Validation, Visualization, Writing – original draft, Writing – review and editing

### Author ORCIDs

Priyadarshan Kinatukara ⓘ http://orcid.org/0000-0003-2210-2369
Sudipta Mondal ⓘ http://orcid.org/0000-0002-3923-7449
Sakshi Shambhavi ⓘ http://orcid.org/0000-0002-8852-1542
Ketan D Patel ⓘ http://orcid.org/0000-0003-4254-3145

Rajan Sankaranarayanan [ORCID] http://orcid.org/0000-0003-4524-9953

**Decision letter and Author response**
Decision letter https://doi.org/10.7554/eLife.70067.sa1
Author response https://doi.org/10.7554/eLife.70067.sa2

## Additional files

### Supplementary files

• Supplementary file 1. List of PDBs of the ANL superfamily family members curated from the RCSB-PDB database.

• Supplementary file 2. A tabulation of the number of atoms from the N-terminal domain of fatty acyl-AMP ligases (FAALs) and fatty acyl/aryl-CoA ligases (FACLs) (excluding hydrogens) at a defined distance from the atoms in multiple conformations of coenzyme A (CoA) seen in the CoA-bound structures of FACLs. The number of atoms of the protein at a clashing distance is an indicator of the space available in the canonical pocket. A higher number of atoms in FAALs indicates the limited space available in the pocket, while the lower number indicates that it is more accommodative in the case of FACLs. Crystal structures of known FACLs have at least one atom.

• Supplementary file 3. A tabulation of well-characterized biosynthetic clusters containing fatty acyl-AMP ligases (FAALs) adjacent to polyketide synthase /nonribosomal peptide synthetase (PKS/NRPS) that synthesize important bioactive metabolites. The phyla of the eubacteria where these clusters are found, the name of the metabolites, the GenBank ID of the biosynthesis cluster, the FAAL (stand-alone or fused) along with their GenBank ID (common name included) and the references where they have been described are detailed.

• Supplementary file 4. A tabulation of the various fatty acyl-AMP ligases (FAAL)-like domains and their domain organizations identified in the different lineages of eukaryotes as per the taxonomic distribution provided at NCBI Taxonomy ( http://www.ncbi.nlm.nih.gov/taxonomy).

• Supplementary file 5. (A) Residues from representative fatty acyl/aryl-CoA ligases (FACLs) (*Hs*FACL, 3EQ6; *Se*ACS, 1PG4; *Mt*FACL13, 3R44; *Af*FACL, 3G7S; *Ec*FACL, homology model from AlphaFold Protein Structure Database) and fatty acyl-AMP ligases (FAALs) (*Mt*FAAL28, 3E53; *Ms*FAAL32, 5ICR; *Ec*FAAL, 3PBK; *Rs*FAAL, homology model generated using MODELLER) at 4.5 Å distance from coenzyme A (CoA) bound to *Se*ACS (PDB: 1PG4) and *Hs*FACL (PDB: 3EQ6) are identified by structural superposition and tabulated. The FAALs and FACLs included here are those that were used for biochemical analysis along with the availability of their crystal structures. The stick representation of CoA (yellow) is shown along the table with entry and active site marked. The variability of residues along the CoA-binding site results in differential orientation of 4'-phosphopantetheine arm during catalysis. For instance, compare the biochemical profiles of F284A/M233A mutation in *Ms*FAAL32 and F265A/M217A in *Rs*FAAL with respect to residues in the vicinity of the residues chosen for mutation. The conserved phenylalanine in FAALs (F284 of *Ms*FAAL32 and F265 in *Rs*FAAL) is flanked by different residues (H288 in *Ms*FAAL32 and A269 in *Rs*FAAL). Similarly, the conserved methionine of FAALs (M233 of *Ms*FAAL32 and M217 of *Rs*FAAL) is abutted by different residues (S314 in *Ms*FAAL32 and A294 in *Rs*FAAL). (B) A tabulation summarizing the biochemical profiles of various mutations generated in the study. Residue numbers for *Ec*FAAL re used as reference in tables for the 'gain of function in FAALs through mutations in canonical pocket' and 'loss of function in FAALs through mutations in alternate pocket,' while *Mt*FACL13 is used as reference in tables for the 'loss of function in FAALs through mutations in canonical pocket.' A tick (✓) mark indicates that the biochemical activity has been performed in accordance with the hypothesis, which is mentioned in the title of each of the table. A hyphen (-) sign indicates that the mutation did not work as per the hypothesis while 'NA' indicates experiment not performed due to protein expression-related problems.

• Supplementary file 6. Primers used for cloning and generating mutants of proteins in this study.

• Transparent reporting form

### Data availability

All data generated or analyzed during this study are included in the manuscript and supporting files. Source data files have been provided for Figure 2, Figure 4, Figure 4—figure supplement 1 and Figure 4—figure supplement 2.

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
