## [Decision Letter]

**Acceptance summary:**

This study addresses the structural basis of the ability of fatty acyl-AMP ligases (FAAL) to exclude condensation of activated fatty acids with coenzyme-A and facilitate the reaction with other 4-phosphopantetheine linked acceptors. This issue is of significant interest with regard to understanding how certain fatty acids are channelled to specific metabolic fates. This work will contribute significantly to our current knowledge of how distinct classes of enzymes divert fatty acids to virulent lipids in mycobacteria, and it will be more broadly of interest for metabolic engineering.

**Decision letter after peer review:**

Thank you for submitting your article "A universal pocket in FAALs ensures redirection of fatty acid pool away from CoA-based activation" for consideration by *eLife*. Your article has been reviewed by 3 peer reviewers, one of whom is a member of our Board of Reviewing Editors, and the evaluation has been overseen by Michael Marletta as the Senior Editor. The following individual involved in review of your submission has agreed to reveal their identity: Satish K. Nair (Reviewer #2).

Essential revisions:

1. In your manuscript you have highlighted a number of residues within the putative binding site for the 4'-pantatheine moieties in the FAAL enzymes that likely preclude the binding of this portion of the substrate. You have subsequently mutated these residues in the FAAL enzymes from three different organisms and have shown in certain instances that the mutated enzymes are now able to functionally activate CoA (Figure 2c). For example, why does the F284A/M233A mutant of MsFAAL32 function so differently than the corresponding mutant of RsFAAL? You need to explain why some of the FAALs behaved differently than others.

2. You have provided further support for the inability of the apparent canonical site in the FAAL enzymes to be functional by mutating the residues within the active sites of certain FACL enzymes to the bulkier ones found in the FAAL enzymes. Many of the constructs resulted in the loss of function and their ability to activate CoA. However, the loss of function was not uniform across the three FACL enzymes chosen. For example, the A276F/A232M mutant of AfFACL is devoid of CoA activity, but the corresponding mutants of MtFACL13 and EcFACL are fully functional. You need to explain why some of the FACLs behaved differently than others.

3. The discussion on the evolution of this system needs to be improved. You should enhance your discussion by looking at the genomic neighborhoods of bacterial FAALs for the presence of PKS/NRPS gene clusters. This analysis would bolster some of the evolutionary claims.

4. It is imperative that the reactions catalyzed by these enzymes be shown as chemical structures for substrates and products (ChemDraw).

5. There are a number of references (see reviewer #3) that should be included.

6. The experimental details for some of the reactions are missing or quite difficult to find. For example, what is the specific fatty acid that is used in your assays?

*Reviewer #1 (Recommendations for the authors):*

The manuscript would have been significantly enhanced if further structural evidence for the newly identified binding pocket in the FAAL enzymes could have been obtained. My suggestion is that the authors use 4-phosphopantetheine itself as an additional substrate for their gain- and loss-of-function mutants and they also use this compound in crystallization trials with the FAAL enzymes to ascertain whether or not the new putative binding site is real or not.

*Reviewer #2 (Recommendations for the authors):*

I would endorse publication pending a stronger analysis to bolster the evolutionary claims made by the authors. With this additional information, this would be a strong publication that would inform across the broad readership of *eLife*.

*Reviewer #3 (Recommendations for the authors):*

I got confused by some of the language, maybe it is just me, the other reviewers may be OK. e.g. Line 46 "they completely lack acceptor promiscuity"….please change this to they display a narrow substrate preference for holo-ACP and not CoASH.

Please start the paper with a chemical, stepwise pathway – e.g. use ChemDraw – to draw out the steps as I described above.

A new review I like which the authors MUST cite is by Winn et al., Harnessing and engineering amide bond forming ligases for the synthesis of amides. Curr. Op. Chem. Biol., 2020, Volume 55, Pages 77-85. This describes the chemistry, mechanism and structures well.

Another to read and cite is Schmelz, S. and Naismith, J.H. Adenylate-forming enzymes. Curr. Opin. Struct. Biol. 19, 666-671 (2009).

The authors should also cite this sequences analysis paper that complements their work.

Clark et al., PLOS One, 2018, In silico analysis of class I adenylate-forming enzymes reveals family and group-specific conservations. doi.org/10.1371/journal.pone.0203218

This combined sequence/structure analysis e.g. STable 1 with all PDBs is very useful.

The data on the mutants were confusing when not related back to the structure of ACPs – they tend to have -ve charge and although the authors discuss CoASH chemistry (e.g. the phosphates) they should also show an ACP – they are always shown as space-filled – show the electrostatics e.g. of EcACP.

There is quite a long conclusion which covers evolution and origin of diversity – this should be edited.

Overall, I think this could be a useful and interesting paper to the field once it is has been edited and over hauled.

Please focus on explaining the biochemical data in a much clearer way.

I suggest accept after major changes.

---

## [Author Response]

Essential revisions:1. In your manuscript you have highlighted a number of residues within the putative binding site for the 4'-pantatheine moieties in the FAAL enzymes that likely preclude the binding of this portion of the substrate. You have subsequently mutated these residues in the FAAL enzymes from three different organisms and have shown in certain instances that the mutated enzymes are now able to functionally activate CoA (Figure 2c). For example, why does the F284A/M233A mutant of MsFAAL32 function so differently than the corresponding mutant of RsFAAL? You need to explain why some of the FAALs behaved differently than others.

We thank you for providing us the opportunity to clarify why there are differences in the performance of different mutants of FAALs.

**Author response image 1. sa2fig1:** A snapshot of residues (stick representation) in FAALs at 4-5 Å distance from Coenzyme A (black colour outlined in red) bound to *Se*ACS (PDB: 1PG4)**.**

**Author response image 2. sa2fig2:** A tabulation of residues in representative FAALs at 4-5 Å distance from Coenzyme A bound to *Se*ACS (PDB: 1PG4). The stick representation of CoA (yellow) is shown along the table with the entry of the pocket and active site marked.

Here, we present a comparative analysis of canonical CoA-binding sites by structural superposition of the N-terminal domains of FAALs (*Mt*FAAL28, 3E53; *Ms*FAAL32, 5ICR; *Ec*FAAL, 3PBK; *Rs*FAAL, homology model generated using MODELLER) with the N-terminal domains of coenzyme A-bound FACLs (*Hs*FACL, 3EQ6 and *Se*ACS, 1PG4). The residues from FAALs in the vicinity (≤ 4.5 Å) of 4'-phosphopantetheine arm, the bound coenzyme A in the crystal structures of *Hs*FACL (3EQ6) and *Se*ACS (1PG4), are tabulated above. The conserved residues are shown in red, while those showing variations are highlighted in yellow.

The residues Phe284 and Met233 were chosen for mutation (marked by asterisk in the table) because of their conserved nature and in the case of FAALs, mutations rendering them smaller is expected to lead to gain of function, i.e., acyl-CoA formation. However, the ability to form acyl-CoA is also dependent on how well the 4'-phosphopantetheine arm is accommodated and poised for catalysis. The effect of variations in residues of the pocket can alter the accommodation of the incoming 4'-phosphopantetheine arm, which is difficult to predict. Such variations of residues in the pocket may possibly allow differential accommodation of 4'-phosphopantetheine arm.

**Author response image 3. sa2fig3:** A snapshot of residues (stick representation) in *Ms*FAAL32 (blue) and *Rs*FAAL (cyan) at 4-5 Å distance from Coenzyme A (black colour outlined in red) bound to *Se*ACS (PDB: 1PG4).

In the case of the specific mutation pointed out by the referee, F284A/M233A in *Ms*FAAL32 (blue) and F265A/M217A in *Rs*FAAL (cyan), the residues adjacent to these sites are those which show variability. For instance, F284 of *Ms*FAAL32 is flanked by H288 (marked by a green circle) while the analogous site in *Rs*FAAL has A269 (marked by a red circle). Similarly, M233 of *Ms*FAAL32 is abutted by S314 (marked by a green asterisk) as against A294 (marked by a red asterisk) abutting M217 in *Rs*FAAL. Therefore, the differences in orientations of 4'-phosphopantetheine arm entering the pocket arising from subtle variations in the pocket could be responsible for the differences in activity of a common site mutation in different FAALs.

We have now added a summary of the explanation on page-12. The tabulation of residues is now included in the supplementary file-5.

2. You have provided further support for the inability of the apparent canonical site in the FAAL enzymes to be functional by mutating the residues within the active sites of certain FACL enzymes to the bulkier ones found in the FAAL enzymes. Many of the constructs resulted in the loss of function and their ability to activate CoA. However, the loss of function was not uniform across the three FACL enzymes chosen. For example, the A276F/A232M mutant of AfFACL is devoid of CoA activity, but the corresponding mutants of MtFACL13 and EcFACL are fully functional. You need to explain why some of the FACLs behaved differently than others.

We thank the referee for pointing out this aspect, and similar to the earlier question, this aspect is also now addressed elaborately.

The likely reason for the differential activity in FACLs is same as described in the earlier question. We performed structural superposition of the N-terminal domains of FACLs (*Hs*FACL, 3EQ6; *Se*ACS, 1PG4; *Mt*FACL13, 3R44; *Af*FACL, 3G7S; *Ec*FACL, homology model from AlphaFold Protein Structure Database) to identify residues from these FACLs in the vicinity (< 4.5 Å) of 4'-phosphopantetheine arm. The residues from different FACLs surrounding the 4'-phosphopantetheine arm in the crystal structures of *Hs*FACL (3EQ6) and *Se*ACS (1PG4), are tabulated in Supplementary file 5A.

The tabulation of residues reveals that unlike FAALs, FACLs have even greater number of divergent residues lining the CoA-binding site. It indicates that accommodation of CoA within the site is likely to be quite different for different FACLs, which is further supported by the different conformations of CoA in the multiple crystal structures of CoA-bound FACLs (Figure 5—Figure Supplement 1a). Thus, the effect of such variations in accommodating the incoming 4'-phosphopantetheine arm possibly plays a role in orienting the CoA differently leading to differential activity.

We have now added a summary of the explanation in page12. The tabulation of residues is now included in supplementary file 5.

3. The discussion on the evolution of this system needs to be improved. You should enhance your discussion by looking at the genomic neighborhoods of bacterial FAALs for the presence of PKS/NRPS gene clusters. This analysis would bolster some of the evolutionary claims.

We appreciate the suggestion of the reviewer, and such an analysis has been carried out by multiple groups for several PKS/NRPS biosynthetic clusters that produce important metabolites. We have collected all the available literature where FAALs have been shown to be important for the biosynthesis of those bioactive metabolites. It is important to note here that not all FAALs from bacteria are in the genomic neighborhoods of a PKS/NRPS cluster. They are sometimes associated with a stand-alone acyl carrier protein domain or fused to other domains such as dehydrogenases, acyl transferases, reductases etc. The known examples of these are seen in *E. coli* and *Legionella pneumophila* (J Mol Biol. 2011;406(2):313-24).

The information is tabulated and included as a supplementary file-3.

4. It is imperative that the reactions catalyzed by these enzymes be shown as chemical structures for substrates and products (ChemDraw).

As suggested by the reviewer, we have now included a schematic of the chemical reactions catalyzed by both FAALs and FACLs with the chemical structures of a generic substrate, different cofactors and products generated using Chemdraw. The same is now presented as Figure 1—Figure supplement-1b.

5. There are a number of references (see reviewer #3) that should be included.

We thank the referee for bringing to our notice that we have missed citing some important literature relevant to the current study. We have now included these as references #10, #1 and # 46.

6. The experimental details for some of the reactions are missing or quite difficult to find. For example, what is the specific fatty acid that is used in your assays?

We thank the referee for pointing this out. The experimental methodology is rewritten with detailing the specific compositions as follows:

Acyl-CoA formation assay: The acyl-CoA formation assay involves separation of the products on a thin-layer chromatography, which are then detected using 1-^14^C labelled fatty acids. 5 μM of enzyme was used for both FAALs and FACLs except *Mx*FAAL and *Rs*FAAL where 7.5 μM was used. The reaction was carried out in a 15 μL reaction volume with 50 mM Tris (pH 8.0), 8 mM MgCl2, 176 μM fatty acid and 24 μM 1-^14^C fatty acid. Lauric acid (C12) was used for all enzymes except MsFAAL32 where palmitic acid (C16) was used as the substrate. 2 mM CoASH was included for assessing the formation of acyl-CoA (product of second step) and omitted in reactions where acyl-AMP (product of first step) was assessed. 2 mM ATP was used to initiate the reaction and incubated at 30°C for 15 min on the water bath. The reaction was quenched using 5 μL of 10% acetic acid. The entire 20 μL reaction mixture was directly spotted on a silica gel 60 F_254_ TLC plate and allowed to dry for 2 hours. Subsequently the products were resolved using n-butanol/acetic acid/water (80:25:40; v/v) solvent system at 4°C. The TLC was quantified by densitometry using Image Lab software (Bio-Rad; version 5.2.1) after scanning the phosphorimage plate exposed for 12-15 hrs. The intensity of each band was quantified as a fraction of total radioactivity in the given lane. Each reaction was in triplicates and the mean for each band intensity was calculated. The percentage of acyl-AMP converted to acyl-CoA was used in the plot.

Acyl-ACP formation assay: The acyl-ACP formation assay involved the identification of the acyl-ACP on a modified radio-Urea PAGE using 1-^14^C labelled fatty acids. The protein concentrations for the pair of FAAL and ACP were 1 μM:20 μM of EcFAAL:EcACP, 5 μM:20 μM of RsFAAL:RsACP and 4 μM:8 μM of MxFAAL:MxACP. The reaction was carried out in a 15 μL reaction volume in 50 mM HEPES (pH 7.2) containing 5 mM MgCl2, 0.01% (w/v) Tween-20, 0.003% DMSO and 9 μM 1-^14^C fatty acid. Lauric acid (C12) was used for all enzymes. In case of MxFAAL and RsFAAL, DMSO was omitted and 50mM Tris (pH 8.0) was used as the buffer in the reaction. 2 mM ATP was used to initiate the reaction and incubated at 30°C for 2 hours on water bath. The reaction was quenched by adding an equal volume of 2X loading buffer containing 25 mM Tris (pH 6.8), 1.25 M urea, 20% glycerol and 0.08% Bromophenol blue. The entire 30 μL reaction mixture was then loaded to 15% PAGE containing 2.5 M urea. In the case of MsFAAL32:MsPKS13∆C, a protein concentration of 1 μM:12.5 μM was used along with all reaction components identical to others except that 1-^14^C palmitic acid (C16) was used. MsFAAL32:MsPKS13∆C reactions were assayed using 8% SDS-PAGE, run at 125 V in the cold room. Gels were dried using gel drier and exposed to phosphor imager screen overnight.

We have rewritten our Materials and methods section to include specific information as indicated.

Reviewer #1 (Recommendations for the authors):The manuscript would have been significantly enhanced if further structural evidence for the newly identified binding pocket in the FAAL enzymes could have been obtained. My suggestion is that the authors use 4-phosphopantetheine itself as an additional substrate for their gain- and loss-of-function mutants and they also use this compound in crystallization trials with the FAAL enzymes to ascertain whether or not the new putative binding site is real or not.

We appreciate the reviewer’s suggestion of attempting a structural characterization of FAALs with a focus on the newly identified pocket. These efforts thus far have unfortunately not yielded any significant success. The indications from biochemical studies will guide us to attempt the structural characterization using different strategies in a future study.

Reviewer #2 (Recommendations for the authors):I would endorse publication pending a stronger analysis to bolster the evolutionary claims made by the authors. With this additional information, this would be a strong publication that would inform across the broad readership of eLife.

We thank the reviewer for the suggestion. The phylogenetic spread and well characterized FAAL containing PKS/NRPS clusters have been compiled and presented as Supplementary file-3.

Reviewer #3 (Recommendations for the authors):I got confused by some of the language, maybe it is just me, the other reviewers may be OK.e.g. Line 46 "they completely lack acceptor promiscuity"….please change this to they display a narrow substrate preference for holo-ACP and not CoASH.

As suggested by the reviewer we have now modified the suggested phrase to “shows a narrow substrate preference for holo-ACP and not CoASH” on page-2.

Please start the paper with a chemical, stepwise pathway – e.g. use ChemDraw – to draw out the steps as I described above.

We thank the reviewer for the suggestion and have now included the reaction schematic, drawn in Chemdraw, as a Figure 1—figure supplement-1b.

A new review I like which the authors MUST cite is by Winn et al., Harnessing and engineering amide bond forming ligases for the synthesis of amides. Curr. Op. Chem. Biol., 2020, Volume 55, Pages 77-85. This describes the chemistry, mechanism and structures well.Another to read and cite is Schmelz, S. and Naismith, J.H. Adenylate-forming enzymes. Curr. Opin. Struct. Biol. 19, 666-671 (2009).The authors should also cite this sequences analysis paper that complements their work.Clark et al., PLOS One, 2018, In silico analysis of class I adenylate-forming enzymes reveals family and group-specific conservations. doi.org/10.1371/journal.pone.0203218

We sincerely thank the reviewer for the pointing out the relevant references. We have now included those as references #10, #1 and # 46.

This combined sequence/structure analysis e.g. STable 1 with all PDBs is very useful.The data on the mutants were confusing when not related back to the structure of ACPs – they tend to have -ve charge and although the authors discuss CoASH chemistry (e.g. the phosphates) they should also show an ACP – they are always shown as space-filled – show the electrostatics e.g. of EcACP.

We thank the reviewer for the suggestion. We agree with reviewer that the ACPs have a negatively charged surface which could dock on the positively charged patch near the canonical binding site as seen in the crystal structures of the A-domain: ACP complexes. However, the absence of the positive patch in FAALs rules out the possibility of such a docking and moreover the canonical site is blocked. Therefore, it becomes difficult to place the 4'-phosphopantetheine in the new site if the ACPs were docked in the canonical site. The approach of ACP against this new pocket is still unknown and requires further investigation. Moreover, the analysis will be further complicated because it is unclear how the C-terminal domain of FAAL is going to be oriented when the ACP docks against this new pocket. Therefore, our submission is that it is extremely difficult to predict the interface residues or mapping the mutations against the ACP in this study. This would require experimental data like structure of the complex etc. which will be a focus of our future work.

There is quite a long conclusion which covers evolution and origin of diversity – this should be edited.

As per the suggestion, we have modified the Discussion section on pages 12, 13 and 14.

Overall, I think this could be a useful and interesting paper to the field once it is has been edited and over hauled.Please focus on explaining the biochemical data in a much clearer way.I suggest accept after major changes.

We thank the reviewer for the observations and suggestions.